# Influence of the COVID-19 Pandemic on Medication Reconciliation in Frail Elderly People at Hospital Discharge: Perception of Healthcare Professionals

**DOI:** 10.3390/ijerph191610348

**Published:** 2022-08-19

**Authors:** María Jesús Rojas-Ocaña, E. Begoña García-Navarro, Sonia García-Navarro, María Eulalia Macías-Colorado, Servando Manuel Baz-Montero, Miriam Araujo-Hernández

**Affiliations:** 1Departamento de Enfermería, Facultad de Enfermería, Universidad de Huelva, Av. de las Fuerzas Armadas, s/n, 21007 Huelva, Spain; 2Distrito Huelva Costa Condado Campiña, Andalusian Health Service, 21700 Huelva, Spain

**Keywords:** medication reconciliation at hospital discharge, COVID-19 pandemic, professional perceptions, polymedicated elderly

## Abstract

The current demographic panorama in Spain corresponds to an aging population; this situation is characterized by the need to care for an elderly population, which contains polymedicated and pluripathological individuals. Polymedication is a criterion of frailty in the elderly and a risk factor for mortality and morbidity due to the increased risk of drug interactions and medication errors. There are numerous studies that measure reconciliation at hospital discharge and at admission, and even the methodology of reconciliation, but we have not found many studies that measure reconciliation in the context of the COVID-19 pandemic from the point of view of health professionals regarding difficulties and the strategies carried out, which is essential to begin to glimpse solutions. Methods. This was a qualitative study based on 21 in-depth interviews and two discussion groups, conducted between January and April 2021 (13 nurses and 8 doctors, in rural and urban areas). The discourse was analyzed according to the Taylor–Bodgan model and processed using Atlas.ti software. Results. The areas altered by the health crisis were access to patients, their reconciliation of medication, and changes in the care modality, including the greater use of telephone communication, changes in work organization, and time dedicated to patient care and family work. Difficulties encountered during COVID-19: change in medication format, the specific characteristics of the patient and their pathologies, and difficulties arising from communication with the patient and their family. The strategies applied: the collaboration of home assistants and caregivers, emphasis on patient–health professional communication, and the use of Information and Communication Technologies (ICT). Conclusion. The discharge was interrupted by the health crisis caused by COVID-19, in terms of both the traditional access of patients and by the remote care modalities generated by telemedicine.

## 1. Introduction

The current demographic panorama in Spain is one of an aging population, with a high proportion of people aged over 65, and a progressive increase in the needs of the over 80 s. According to data from INE (the Spanish government’s national statistics institute), as of 1 January 2019, there were 9,057,193 elderly people, representing 19.3% of the total population, of which 6.1% were octogenarians, a segment that will continue to grow and gain in significance as the overall population ages [1].

This situation is characterized by the need to attend to a population with chronic diseases that require follow-up and the control of symptoms, which entails increasing pressure on health services and which results in an aging, polymedicated population. Polymedication is a criterion of frailty in the elderly and a risk factor for mortality and morbidity due to the increased risk of drug interactions, adverse effects (risk of falls, cognitive impairment, and sensory alterations), decreased therapeutic adherence, and greater use of health resources [2].

The strategy for addressing chronicity in the national health system [3] establishes objectives and recommendations that guide the organization of services towards improving the health of the population through comprehensive care, which includes pharmacological assessment.

In this sense, the Integrated Healthcare Process for Multipathological Patient Care (PAI-APP) [4], developed by the Junta de Andalucia (Spain), has tried to transcend the perspective of classic integrated geriatric assessment by incorporating what is known as the Exhaustive Integrated Assessment (EIA), which covers clinical, functional, cognitive, affective and socio-family assessment, prognostic and pharmacological assessments, the assessment of self-management capacity and a person’s activation in the management of their disease, and the care and control of medication.

The National Coordinating Council for Medication Error Reporting and Prevention (NCC MERP) defines medication errors (ME) as “any foreseeable incident that may cause harm to the patient or result in the inappropriate use of medications” [5]. Likewise, the National Institute for Health and Care Excellence guide defines medication in their 2008 review as “critically and structurally assessing a patient’s medications in order to achieve an agreement with the patient on treatment, optimizing the impact of medications, minimizing the number of drug-related problems (DRP) and reducing costs” [6].

Medication errors are common in the hospital [7] setting and lead to increased morbidity, mortality, and economic costs. Most of the effects occur at hospital admission and/or on discharge, that is, in the transition of patients between different levels of care [8,9,10]. These errors are due to the total or partial ignorance of a patient’s current treatment, in which necessary outpatient medications are omitted, duplications occur, and incorrect doses are prescribed, etc. [11].

A key element in preventing or avoiding these medication errors is treatment reconciliation, that is, coordination between different healthcare areas and professionals.

Treatment reconciliation is the formal process of assessing the complete, accurate list of the patient’s previous medication matched to the pharmacotherapeutic prescription is-sued after a care transition, admission to hospital, a change in medical manager or hospital discharge [12,13]

However, the literature consulted for this study suggests that errors in medication reconciliation are more frequent at hospital discharge [14,15,16,17]; thus, the process of medication reconciliation needs to form part of the daily healthcare routine in all hospitals, since it has proven its worth in reducing medication errors and their potential consequences for the elderly [14,17,18,19].

Another element that would reduce medication errors is the adoption by healthcare professionals of the criteria for potentially inappropriate medications/potential prescribing omissions in older people (STOPP/START) [20].

Equally, we cannot conduct this process without considering the characteristics of the elderly patient, which encompass the physiological changes typical of the aging process, as well as changes in drug pharmacokinetics and pharmacodynamics that make the elderly especially sensitive to the consequences of medication errors of a more serious nature than the rest of the population [21,22].

These changes increase considerably with age, increasing the risks associated with polymedication, which can, in turn, lead to more medication errors in elderly patients [23]. From the age of 75, the rate of patients with chronic diseases increases, as does their attendance by healthcare professionals, which can involve more polymedication, a greater risk of adverse drug reactions, poor compliance, and increased hospital admissions [24,25,26].

How has the COVID-19 pandemic influenced this process?

Without a doubt, the pandemic has marked a before and after both at the organizational level in health centers and in the way of attending to the demands and needs of patients, as well as in the performance of primary care professionals. It has also affected drug-related issues and the medication reconciliation process [27]. COVID-19 has shaken the health system in many ways, especially primary care.

Healthcare has gone from being administered face-to-face to telephone consultation, in order to ensure the safety of patients and healthcare professionals while continuing to try to satisfy patients’ demands and needs.

Prior to the pandemic, numerous studies analyzed medication reconciliation at hospital discharge [26,28,29,30,31] and at admission [32,33], and even the medication reconciliation methodology itself [20]. However, post-pandemic studies have focused primarily on patient perspectives on telemedicine [34,35,36] and the barriers to patient access of the new forms of healthcare, such as low technological competence [37].

However, probably due to the recent nature of the COVID-19 phenomenon, we have found no studies that measure medication reconciliation in the context of the pandemic from the point of view of healthcare professionals, their perceptions, the difficulties they face, and the strategies developed and performed in their daily practice.

Therefore, the main objective of this study was to know the perception of primary care professionals of the influence of the COVID-19 pandemic on the medication recon-ciliation process in patients over 65 years of age after hospital discharge. Secondary objectives included the analysis of the difficulties encountered and the strategies applied to medication reconciliation, as well primary care professionals’ proposals for improvements to the medication reconciliation strategies.

## 2. Materials and Methods

The methodology applied in this study was of a qualitative phenomenological nature, given our interest in the emic approach to the study subjects. This research adheres to COREQ guidelines [38]. Our analysis was based on information gathered from 21 in-depth interviews and two discussion groups staged between January and April 2021, with the aim of knowing the influence of COVID-19 on medication reconciliation. Focusing on this dimension in the interviews enabled us to analyze in greater depth the plots that were subsequently superimposed on the discourses generated in the two discussion groups, and with that, the ideas that condition specific ways of acting, facilitating an understanding of difficulties and strategies. The arguments used to understand each of the reactions of the primary care professionals during the social context generated by the pandemic represent the tools by which each of the actors justifies their position and articulates their self-generated discursive strategy to the rest of society. Thus, an inductive analysis of the arguments and the affinity relationships between them was used to obtain the main arguments deployed in the context of this study phenomenon.

### 2.1. Participants and Study Environments

The sampling strategy was intentional sampling, relying on the validity and reliability of the information that the selected population can provide [39] and evidence of similar studies [40,41]. The study population was considered from the range of healthcare professionals (doctors and nurses from primary care) in relation to medication reconciliation in patients over 65 who worked in rural and urban health centers during the COVID-19 pandemic.

The sample size was determined progressively during the course of incorporating research informants until reaching the saturation of the information [42]. The inclusion criteria considered were the same for both groups of professionals, nurses and doctors, having a minimum of 2 years of professional experience in the field of primary care and having been active during the pandemic period, in addition to providing verbal and written consent to voluntarily participate in the research. Geographical variables of the work environment were taken into account, trying to achieve the maximum representativeness of the sample in terms of the discourses of professionals who worked in rural and urban health centers.

The procedure that was carried out to recruit key informants was through the coordinators of the centers, as well as the quality director of the health district, who is a member of our research team. This member served as a liaison with potential candidates in order to select subjects with experience in the object of study. In turn, due to the greater difficulty of access to potential participants derived from the accumulation of professional tasks and geographical dispersion, we resorted to the snowball sampling method [43]. The participants were contacted by phone and, after verifying that they met the agreed criteria to participate voluntarily as informants, they were summoned for an interview, with this main question: What is your opinion regarding the reconciliation of medication in people over 65 years of age during the COVID-19 pandemic and what are the factors that have influenced it?

In order to obtain sufficient variability and provide greater depth to the discourse analysis, a number of structural variables were taken into account concerning the choice of participants. Specifically, the professionals were of different ages, with different years of professional experience, and all belonged to primary care centers in different geographical areas.

Our final selection was 13 nurses and 8 doctors, of whom 8 belonged to rural areas and 13 to urban areas, obtaining a total of 21 interviews.

Study period: Fieldwork was carried out between January and April 2021.

The interview script covered dimensions associated with the location in which the primary care professional works, their professional career, the type of healthcare activities performed in their regular primary care practice, the particularities of the geographical environment where they work, the tools and practices applied in medication reconciliation before a patient is admitted to hospital, the medication reconciliation procedures that follow admission, the changes caused by the COVID-19 health crisis in their usual healthcare practice, the perceived difficulties in medication reconciliation, and the influence of COVID-19 on medication reconciliation and the functioning of health services.

The in-depth interviews were conducted in online format through the Zoom^®^ Platform, thus facilitating accessibility to the informant given their geographical dispersion.

### 2.2. Analysis

We analyzed the discourse in line with the phenomenological paradigm inspired by repetitive “decontextualization” and “recontextualization” processes, in order to understand the experiences and perspectives of the interviewees following the COREQ criteria regarding qualitative analysis [38]. This model differentiated three different phases in the processing of the data collected. First was the action or discovery phase, which focused mainly on data collection and reiterated the reading of the interviews in order to extract the most repeated topics; this enabled us to obtain different categories and develop related theoretical concepts. Once completed, a new reading of the bibliographic content related to the topic used in this study was carried out to develop an interview guide

The next phase was coding. With the different categories and emerging topics in the interviews now identified, we proceeded to separate the various data obtained. According to the categories established, only the most useful data were selected to help us refine the analysis and extract the greatest potential from the interviews.

Finally, the “data relativization” phase was carried out, which not only took into account the theoretical data extracted in the interviews, but also variables such as whether or not these interviews were requested, how our presence could affect the conduct of the interview, the characteristics of the environment in which the interviews took place, and the interviewer’s (the authors) own assumptions.

We used the Atlas.ti data management program for the coding and recoding process, to identify all the arguments expressed by the interviewees. This program quantifies the citations for each of the codes assigned during the analysis of the interviews.

### 2.3. Ethical Considerations

The study was approved by the ethical health research committee of the province of Huelva, Spain, which was assigned code PPPCM/21. Verbal consent was granted by the participants in the interviews and focus group discussions. Throughout the interview process, the unintended consequences of interviews and focus group discussions were always taken into consideration. The data were processed in accordance with Organic Law 15/1999, of December 13, on the protection of personal data (BOE 298 of 14 December 1999). The study was carried out ensuring compliance with current ethical and legal standards (Declaration of Helsinki).

## 3. Results

### 3.1. Participants’ Characteristics

The data were extracted from the information collected from 21 interviewees (13 nurses and 8 doctors), from different primary care areas, and rural and urban settings. Of the 21 subjects interviewed, 6 nurses worked in a rural setting, 7 nurses in an urban setting, 3 doctors in a rural setting, and 5 doctors in an urban setting (Table 1). The average number of years’ experience as a health care professional was 25.76 years, with 15.59 years dedicated to primary care. The health care professional with the shortest time in primary care had 2 years’ experience, and the longest time served in primary care was 27 years.

The distribution by sex in the study was 83.3% women and 16.7% men; this is due to the fact that women represent almost 65% of the total nursing profession. The average age of the participants was 50.57 years, with 24.14 years’ experience in primary care. The sample consisted of 61.90% nurses and 38.10% doctors (Table 1).

The coding of the interviewees’ arguments generated 41 codes, divided into four groups from which those related to the study objectives were selected: influence of the pandemic (11), difficulties in medication reconciliation (8) and reconciliation strategies used (8) (Figure 1).

The analysis of principal components revealed three lines of argument that conveyed the participants’ discourse, coinciding with the objectives of this study (Table 2).

### 3.2. Influence of the COVID-19 Pandemic on the Primary Care Practice Performed by Nurses and Doctors

The discourse analysis provided 11 codes. The areas altered by the healthcare crisis were access to patients and their medication reconciliation. The analysis also revealed dis-crimination in terms of the healthcare provided and the cases attended, uncertainty due to ignorance of the virus, and changes to the mode of healthcare they were used to dispensing: a greater use of telephone communication, changes in the organization of work, and time dedicated to attention to the patient and to work with the family in the process. The discourses also revealed a change in the vision of primary care work, with the perception of greater empowerment across the wide range of primary care activities (Table 3).

### 3.3. Difficulties Encountered during the COVID-19 Pandemic in Medication Reconciliation after Discharge in Patients over 65 Years of Age

This line of argument presents eight codes that correspond to crucial issues in medication reconciliation, such as: a change in medication format, the specific characteristics of the patient and their pathologies, and difficulties arising from communication with patients and family (Table 4).

The limited specific training that healthcare professionals receive in medication reconciliation issues, workforce stability, and the reduction in the time dedicated to each case represent other codes that figure in the analysis of this study. Finally, nurses and doctors pointed to the difficulties resulting from a lack of common criteria in the control of medication and, therefore, in the choice of medication offered to the patient.

### 3.4. Strategies Applied by Healthcare Professionals during the Pandemic to Medication Recon-Ciliation

This dimension generated nine codes, which encompass the collaboration of home-help assistants and caregivers, an emphasis on patient–healthcare professional communication, and the use of Information and Communication Technologies (ICT). This line of argument also refers to the tasks developed to promote trust between the healthcare professional, patient, and family. Telephone consultations have been a key element in ensuring correct medication reconciliation, as well as the use of specific tools, graphic schemes, pillboxes, and face-to-face visits in cases of need (Table 5).

## 4. Discussion

The results obtained in our study reflect the significant influence of the COVID-19 pandemic, resulting in changes to the healthcare system and, of course, to primary care [27]. A range of actions have been developed, prioritizing the diagnosis and care of patients with COVID-19 over second-line actions previously carried out in-person [44]. In this sense, telemedicine has taken on a great role in all areas of healthcare [45,46,47], in some ways exacerbating existing barriers between the healthcare system and patients and families; in short, both doctors and nurses have had to confront a new disease and develop a new way of interacting with patients.

As the results show, this situation has generated difficulties in different groups, such as frail elderly people in the telephone follow-up of consultations, due to ignorance of new technologies or linguistic or cognitive barriers [48,49,50]. In line with these results, Lang [51] reported that patients over 85 years of age of low socioeconomic status may be particularly affected by this digital divide, which has been widened by the pandemic.

Telemedicine has cut the personal contact between healthcare worker and patient, reduced accessibility, and affected patient treatment, essential elements in the development of the healthcare worker–patient relationship; it was also indicated as a difficulty in the medication reconciliation process by our participants [52]. However, authors such as Keesara et al. [53] state that telemedicine can be an advantage in the practice of medication reconciliation, as long as patients can connect by camera and can visualize changes in the medication format, as well as read the labels.

However, overall, and following the thread of difficulties encountered by the study participants, it is communication that is fundamental, as it was before and now in the midst of the pandemic, not only across all levels of healthcare [17,54], but also between healthcare personnel, their patients, and the families. Our results show that communication is an essential nursing skill. Face-to-face communication is vital, enabling nurse and patient to have a full, unimpeded conversation where both parties speak and listen, in which the healthcare worker can also register body language and the facial expressions that provide key information to enable them to better understand the meaning of what they hear from their patient [53].

Regarding medication reconciliation strategies, the findings point again to the need for communication effectiveness in regard to pharmacotherapeutic information among health professionals [55,56,57], the lack of coordination between the different levels of care, or the absence of a single record of the patient’s medication, which must be correct, updated, and accessible [23,58]. Other authors focus on the training of health professionals [59] and the close collaboration with family members or home help assistants to cover the basic needs of the elderly, including the control of their medication [52,60,61].

There are studies at the international level and across all healthcare settings that emphasize the need for interventions to improve prescription, to eliminate medication-related problems, and to ensure the suitability of the medication prescribed [15,16,18].

In Spain, the EARCAS study of adverse events in nursing homes and socio-sanitary care centers called attention to the problem of drug safety [19]. A formative intervention carried out with primary care physicians was associated with a reduction in polymedication of more than 10 drugs, and in inappropriate prescription under typical clinical practice conditions [62], and even obviated reconciliation programs upon hospital discharge [17].

As already mentioned, medication management is the most complex component in the patient’s transition from hospital to home after discharge. Kollerup [63] showed how a good nurse–patient relationship developed following hospital discharge, enabling good medication reconciliation, and helping the patient to perform their own safe medication administration.

Another aspect of this transition from hospital environment to primary care is improving patient safety by facilitating communication and coordination between professionals through the use of digital tools that enable the transfer of information; this ensures continuity of care for patients, especially in the case of elderly, polymedicated, multipathological patients [64].

Therefore, for a medication reconciliation program to have a positive impact on the quality of pharmacotherapy, a multidisciplinary approach is essential [54,65]. This re-quires the different healthcare professionals involved in an elderly patient’s care to share responsibility. Primary care and internal medicine physicians play a notable role in this process [30,66,67], due to the profile of patients they usually attend, namely, general patients with multiple risk factors associated with the appearance of unintentional discrepancies, such as polypharmacy, comorbidities, and, in many cases, with difficulties in communication due to age or clinical situation.

In this sense, patient safety was found in the study conducted by Rico [68] to be the greatest concern during the pandemic, according to the nurses’ perception, when per-forming medication administration procedures, in addition to an overload of work, insufficient training, and, again, failures of communication and information.

On the other hand, the results highlight the need to develop strategies that empower patients and their families, in order to improve said safety [69], in addition to giving patients greater confidence to manage their own medical care and resolve health issues.

This involves sharpening patient’s healthcare skills and changing daily routines, the correct use of pillboxes, and calendars or reminder systems [70].

Finally, the primary care approach to COVID-19 has entailed long hours of work that would normally be dedicated to other services provided by primary care professionals [71]

Attention to regular patients has diminished in the frontline fight against the pandemic, and when we return to normality, we will need to analyze the impact these changes have made and how to address them.

## 5. Limitations

As limitations, the difficulties encountered in the bibliographic search for studies on medication reconciliation in the elderly in the context of the pandemic were obviously due to the novelty of the phenomenon. However, we have been able to contrast our results with studies that highlight the importance of medication reconciliation and communication between all levels of the healthcare system, and with patients and families, among others. Our search also found studies that emphasize the role of new information and communication technologies (ICT) in the socio-sanitary setting, which have developed rapidly during the pandemic.

Such limitations should encourage more research to complement our findings, and to enable us to establish future care strategies to guarantee safe medication reconciliation for elderly patients discharged from hospital.

Another limitation was the fact that the interviews were conducted during the COVID-19 pandemic, so they were virtual, preventing face-to-face interaction and the creation of a climate of trust.

### Recommendations for Practice

Health professionals should pay special attention the transition from the hospital environment to primary care in order to improve patient safety. It is necessary to facilitate communication and coordination between professionals through the use of digital tools that allow the transfer of information.

This ensures continuity of care for patients, especially in the case of polymedicated and multipathological elderly patients.

## 6. Conclusions

According to the perception of primary care professionals, medication reconciliation in the elderly following hospital discharge has been disrupted by the health crisis caused by COVID-19, mainly in terms of traditional patient access and as a result of remote modes of care generated by telemedicine.

Regarding the difficulties encountered in the medication reconciliation process, the change in the medication format, the specific characteristics of the patient, the problems involved in communication with patients and the family, and those between different levels of healthcare stand out.

The strategies applied to these new contexts have been a priority for healthcare personnel, with a greater understanding of the need for collaboration between home-help assistants and caregivers, a greater emphasis on patient–healthcare professional communication, and the improved use of ICT. The results of our study also show the importance of promoting trust between the healthcare professional, patient, and family.

Finally, for these actions to be performed successfully requires coordination between healthcare professionals, such as medical and nursing staff, in both hospital and primary care settings, and it will be fundamental to maintain these collaborative working practices in the future. For this to happen, it is necessary to improve communication between the different levels of healthcare practice and among healthcare professionals, in order to inform professionals quickly and efficiently of any failure in therapeutic adherence or of a problem related to the elderly patient’s medication reconciliation.

## Figures and Tables

**Figure 1 ijerph-19-10348-f001:**
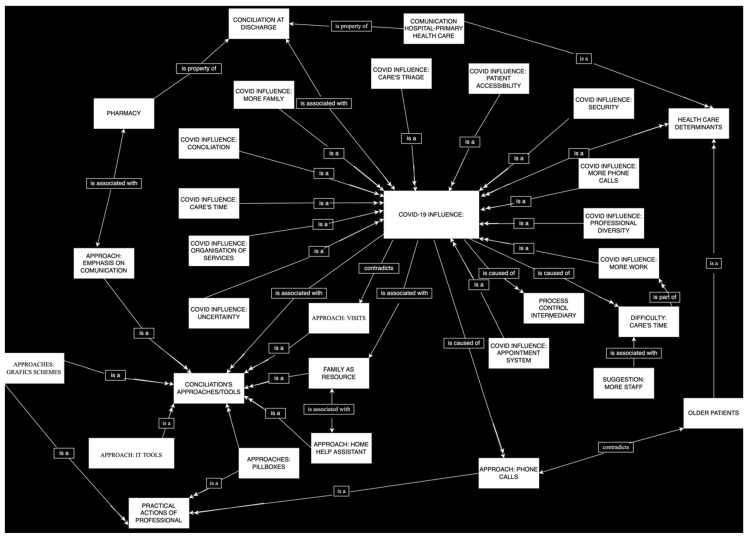
Codes relating to COVID-19 influence on medication reconciliation.

**Table 1 ijerph-19-10348-t001:** Characteristics of participants.

Sex	Profession	Age	Years of Experience	Ambit
F	Nurse	30	6	Urban
F	Nurse	53	12	Urban
F	Nurse	53	19	Urban
F	Nurse	50	20	Rural
F	Nurse	49	27	Urban
F	Nurse	55	30	Rural
F	Nurse	51	30	Rural
F	Nurse	54	32	Urban
F	Nurse	58	34	Rural
F	Nurse	62	38	Urban
M	Nurse	42	17	Urban
M	Nurse	56	31	Rural
M	Nurse	58	35	Rural
F	Doctor	33	8	Urban
F	Doctor	35	11	Urban
F	Doctor	54	25	Urban
M	Doctor	57	21	Rural
M	Doctor	50	25	Urban
M	Doctor	52	27	Urban
M	Doctor	53	28	Urban
M	Doctor	57	31	Rural

**Table 2 ijerph-19-10348-t002:** Description of the categories, codes, and number of citations in the Atlas.ti analysis.

Storyline	Codes	Dating
COVID-19 Influence	Patient accessibility	18
	Conciliation	52
	Discrimination care	6
	Empowerment of primary care	1
	Heterogeneity of professionals	2
	Uncertainty	5
	More family	7
	More work	16
	More phone	41
	Organization of services	3
	Attention time	2
Difficulties	Drug format change	2
	Patient-specific characteristics	7
	Patient communication	4
	Little conciliation training	1
	Unstable template	3
	Attention time	6
	Conciliation control	30
	Medication control	70
Strategies used for conciliation	Auxiliar of home help	14
	Emphasis on communication	5
	Herralies Computer Science	5
	Promotion of trust	4
	Conciliation tools	60
	Using graphical schematics	4
	Pillboxes	2
	Visits	2
	Review medicine cabinets	15

**Table 3 ijerph-19-10348-t003:** Description of the influence of COVID-19. Codes and discourses of the nurses and primary care doctors.

Storyline	Code	Dating	Role	Discourse
Influence of COVID-19	Patient accessibility	18	Nurse	*“There was a sense of abandonment from the patients. When you give those people confidence, it’s not that they abuse consultations, they’re responsible and they really only go when they need to. That gave them a lot of security, knowing that you are there for when they need it, both at home and in the health center. They have not had that feeling during the pandemic, and they have been more reluctant to contact us.”*
			Doctor	*“Yes, totally, especially for having restricted physical contact with the patient. The overload that has occurred with the attention to Covid cases has made it very difficult, not only for the process of medication reconciliation, but also for many other health programs. Even patients with decompensation in their pathology are often afraid to go to the center for fear of contagion, and the situation becomes complicated.”*
	Medication reconciliation	52	Nurse	*“Yes, apart from that, we also began telephone consultations, but these do not work well with the elderly, sometimes they have hearing problems. You can’t check the medicine cabinet. Diabetics show worse control; we have to call them to ask them about the controls, everything is much more difficult for them. For both us and the doctors, with telephone consultations contact with the patient is much diminished, and trust is not fostered.”*
			Doctor	*“It affects everything. We have sometimes been overwhelmed in our attempt to manage resources, and it is true that we have focused on more important issues. This is what should be explained to the patient by the hospital: they give the patient the report on discharge but they do not explain anything; if a patient comes to us with an explanation from the hospital, this information should also be provided to us. When you are overloaded and the patient comes to you with something that has been generated by another specialist, another partner, because the truth is that yes, it affects you, then it overloads you even more. That is what I said, in general there is not that connection or communication that there should be in the relationship between hospital management and primary care.”*
	Care discrimination	6	Nurse	*“In this area you are seeing more people, but not like before, far from it. The pandemic has made everything worse for the chronically ill.”*
			Doctor	
	Primary care empowerment	1	Nurse	
			Doctor	*“Primary care has been the wall of containment of the disease but without the impact that hospitals have had, as 95% of patients have been treated in primary care, for everything, both Covid and non-Covid. Serious cases, admissions, deaths arrive at hospitals, but the vast majority of diagnoses, treatments and follow-ups are done by us.”*
	Professional heterogeneity	2	Nurse	*“Now with the pandemic, some centers have adopted a unified appointments diary in which all the patients are entered, and as they are arrive, they are attended to by the first nurse who is free; in this way, a person is sometimes attended to by different nurses on successive days, which can lead to a lack of coordination in the treatment. I notice this particularly in the curing treatment, in which each nurse has their own criteria. It can be supervised by another, but I believe that curing treatment should always be performed by the same nurse.”*
			Doctor	
	Uncertainty	5	Nurse	*“We were a little nervous at first, since we did not know how the virus was transmitted, and you took many measures; now we are a little calmer. We work with FP2, we do daily antigen tests whenever there is any suspicion of Covid symptoms.”*
			Nurse	
	More family involvement	7	Nurse	*“Home visits were previously made on alternate days; we were concerned about instructing families to go only once a week, and the rest of the days the home help takes care of the patient.”*
			Doctor	
	More work	16	Nurse	*“The pandemic has affected us; since we now work in another way, many patients are not coming to the center. We have other tasks that we did not have before, and we have to take into account that patients continue to get sick from other pathologies, but now we are focused on Covid.”*
			Doctor	*“I know that in Diraya there is a medication reconciliation program, I have used it, but not lately; we are very overwhelmed now with the pandemic.”*
	More hone consultations	41	Nurse	*“Now it’s all about medical consultations by phone. But in our case, 99% of our visits are in-person, which is the opposite for the doctors, 90% of whose consultations are by phone; only 10% face-to-face, though this is increasing little by little. Patients can be confused by phone consultations; it’s not the same as a consultation in-person where you can say things more clearly and directly. But it’s true: patients talk more by phone to the doctor than to us. What is clear is that there are many cases of chronic patients who have not taken their medication correctly, or others who now have up and down blood sugar levels who used to maintain a much steadier line before because now they cannot do physical exercise. These patients have become much more unstable during the pandemic.”*
			Doctor	*“Yes, of course, all these changes in the form of patient care, more telephone than face-to-face, have further complicated the issue of medication reconciliation. We have had to increase vigilance, be more on top of patients who, in many cases, had difficulty accessing their doctor in any other way than the telephone, and sometimes this was the only solution: tell the patient or caregiver to come to the consultation with their bag and list of medicines, then review them with them giving them instructions and clarifying their doubts personally. In the area where I am, which is for older patients and who are, in general, the ones who come here the most, we try to resolve administrative issues by phone consultation, but everything related to medication, because the truth is that it makes it a little difficult, many patients tell me that they prefer to come to the consultation in-person to explain; they have to make a great effort to do so. If, with the patient sitting in front of you, it is difficult for them to understand, then on the phone even more. Now I cannot write things down for them and hand them the piece of paper; sometimes I leave the paper on the counter and tell him to pick it up, but it’s hard to know how the patient will interpret it when he is alone. The restriction on face-to-face consultations, and especially for older people, makes it more difficult for them to understand, quite apart from the fact that people like face-to-face consultations more.”*
	Organization of services	3	Nurse	
			Doctor	*“We are eight doctors, then there is a duty doctor for discharges who covers hospital departures and administrative permits,* etc. *Then there are the rotating doctors who do the emergencies. Before Covid, the emergency room began its work at 20:00, with one doctor. We had morning and afternoon consultations, I had two afternoons and three mornings, and the rotating doctors worked every four days, from 20:00 to 08:00. Since Covid, rotation has been adapted to start at 15:00, and there are two teams of two doctors and two nurses; in the rotation teams there are four doctors.”*
	Patient attention time	2	Nurse	
			Doctor	*“I would like us to devote more time to patients, it would be ideal and patients would be better cared for, and we can solve all their doubts. Keep in mind that not all patients need the same consultation time, and not everyone assimilates the information at the same speed.”*

**Table 4 ijerph-19-10348-t004:** Description of the difficulties involved in medication reconciliation. Codes and discourse of nurses and primary care doctors.

Storyline	Code	Dating	Role	Discourse
Difficulties	Drug format change	2	Nurse	*“Sometimes, it is people’s level of education, not all cases are the same. An important handicap is the changes to generic drugs; sometimes, as I told you, we have people who were taking double the medication thinking they were different drugs.”*
			Doctor	*“I would like to comment on the issue of prescriptions by active ingredient in the elderly. In many cases, the brand or laboratory that supplies them can vary according to when the temporary purchase is made, so the container or the tablet itself may vary, and that leads in many cases to confusion in patients. I have had patients who have stopped taking their antihypertensive medication, or antidiabetics because it is not the pill they were taking before. On many occasions in the pharmacy, they do not stop to explain that it is the same as what they were taking before and that only the brand has changed. For all these reasons, I think that the prescription for the active ingredient is, of course, necessary, but I think it should be modified; for example, the same active ingredients should have a similar packaging, and that there are no significant differences between them, so that they can facilitate understanding. Bear in mind that we have older patients, many with sensory difficulties and who sometimes have a hard time identifying medications.”*
	Patient-specific characteristics	7	Nurse	*“A person’s cognitive level is also influential; people who start to suffer memory lapses, or Alzheimer’s. That’s why we are more on top of these people, and also their relatives, to make them more alert to them taking the medication correctly.”*
			Doctor	
	Patient communication	4	Nurse	
			Doctor	*“Among the biggest drawbacks that I find, and why I have had to call patients and sit them down in the consultation and explain the medications one by one, is the way in which they are given information on medication. I receive a discharge report, but sometimes, and especially in elderly and polymedicated people with memory deficits and so on, they can confuse the medications they have to take. I think the origin of all this is that somebody tells them very quickly what they need to do, they give them a paper, a document, that they tell them to show the doctor, but when you talk to them, you see that they are not taking the medicines correctly; then you have to sit them down and take a little time to go through each of the boxes and explain how they have to take the medicines. I’ve found that on several occasions.”*
	Lack of medication reconciliation training	1	Nurse	*“Over the years, I have done many primary care courses, some of them deal with medication in chronic patients, how to help patients with their medication, etc., but nothing specifically on medication reconciliation.”*
			Doctor	
	Work stability	3	Nurse	*“The lack of stability in the hiring of doctors results in a big turnover, and this has a big influence. Nurses are in the same situation.”*
			Doctor	*“Another handicap we are having to endure is that there have been many changes of doctor; there is no stability in the roster of physicians.”*
	Patient attention time	6	Nurse	*“Yes, but there is no time, time is very limited, I have 15 min in consultation to instruct a patient on how to inject insulin for the first time, for example; that is ridiculous, and you know that if you take longer, you have less time to make an outside visit.”*
			Doctor	*“I would like us to have more time to devote to patients, it would be ideal and patients would be better cared for, we can solve all their doubts. Bear in mind that not all patients need the same consultation time, and not all assimilate the information at the same speed.”*
	Medication reconciliation	30	Nurse	*“If you look, the portfolio of services we offer is growing, more and more things. And it is true that performing all these things excites me, but there comes a time when you have no material space and you need more hours in the day to carry out everything you want to do. Therefore, I think it is a matter of time and personal resources.”*
			Doctor	*“The other thing I wanted to comment on is the feeling, since I have no data yet, that medication reconciliation by phone is extremely complicated; this is done much better in person at the consultation room or at the patient’s home. I have the feeling when I try to reconcile by phone that I am not doing things well; it is more complex, unless your interlocutor is young, with a certain level of education, or a very savvy person.”*
	Medication control	70	Nurse	*“I think the same about phone consultation; the patient tells the doctor that they have understood, then they come to you, and it happens that sometimes we have had many changes of doctor, little staff stability, so they come to us, as nurses are a more stable presence, to ask us, since they were in a hurry to go to the doctor who they did not know. They had more confidence in us and asked us more questions to resolve their doubts. Then, you realize that they did not understand the things that were being explained to them by the doctor.”*
			Doctor	*“One factor is the old age of the population. This complicates explanations about treatments, medication, etc.”*

**Table 5 ijerph-19-10348-t005:** Strategies used for drug reconciliation in the COVID-19 health crisis. Codes and discourse of nurses and primary care doctors.

Storyline	Code	Dating	Role	Discourse
Strategies for medication reconciliation	Auxiliary help at home address	14	Nurse	*“For those patients who are immobilized, and with significant cognitive impairment, it is caregivers, both formal and informal who provide the care. About 3% of patients have a caregiver 24 h a day. These people have been trained and are experienced, they are my eyes, as is the company that provides the service, and even the relatives who are in their charge. Everyone helps me a lot when it comes to assessing any error or problem, and they tell me directly, in the moment.”*
			Doctor	*“We also insist that they bring the bag with the medicines. I have to say that we have a lot of support from the home-help assistants, with whom we have a lot of contact, and they help us a lot on the subject of medication. In the two towns where I work, this is a fundamental help since there is a large population of over 80 s, and in many cases they do not have any family support, many cannot read; fortunately, we have this help.”*
	Emphasis on communication	5	Nurse	
			Doctor	*“The first thing is to improve inter-level communication, especially between primary and specialized care; it needs to be more fluid, without the big communication barrier that we have had; it is true that things are more fluid now, although there is still room for improvement. If we could achieve that, then the patient who is being seen by a specialist colleague of mine, and by me, in the more integrated medical field, is likely to be much better treated and, above all, more closely controlled in terms of medication reconciliation.”*
	Digital tools	5	Nurse	*“First there was the Care Continuity Report, which lay the first stone, and the famous Continuity Book, then Diraya came along and dismantled all that. Today care continuity is guaranteed as long as there is interest on both sides, that is, if you are a professional, you are obliged to use the mailbox since it is your patient who is writing. I always say that Diraya is there, but those records need to contain quality information; you have to specify the actions, symptoms, treatments, attitudes, etc.”*
			Doctor	*“I think the experience I have had is good, since the closeness, the Diraya tools we use, and the communication with the specialists and others, all this is allowing us to carry out much better medication reconciliation for the elderly than before. So much has changed since when I started working.”*
	Promotion of trust	4	Nurse	*“It’s not only that the phone is used all the time, but because of the confidence that there was before, in which the patient could approach you in the health center and ask you all kinds of questions; now I have to tell them that I cannot attend to them, that I will visit that day. That freedom to go to the health center when they wanted has been lost. This is particularly so in the case of chronic patients, so many of whom are afraid to leave their homes.”*
			Doctor	
	Skills conciliation	60	Nurse	*“The strategy would be, as we have said, to design a protocol especially for hospital discharges, which might involve prescribing new medication that departing patients are not very familiar with.”*
			Doctor	*“I think the experience I have had is good, since the closeness, the Diraya tools we use, and the communication with the specialists and others, all this is allowing us to carry out much better medication reconciliation for the elderly than before. So much has changed since I started working.”*
	Using graphic schemes	4	Nurse	*“With older people who have been discharged from hospital with new treatments that they are not familiar with, I try to explain to them what the pill is, how to distinguish it (since there are people who do not know how to read), what it is for and when it has to be taken. Sometimes I make a drawing. If they have caregivers, I explain it to them as well. The pillboxes are also very useful; they divide the pills into sections of the day when they have to take them, and when they are pending. There are also patients who are familiar with one laboratory’s pills, then the laboratory changes the format and messes it up. I want to emphasize this point, since it could be that the patient has the old version of the pills, and what I try to do is avoid them taking the same pill twice.”*
			Doctor	
	Pillboxes	2	Nurse	*“The pillboxes are also very useful; they divide the pills into sections of the day when they have to take them, and when they are pending. There are also patients who are familiar with one laboratory’s pills, then the laboratory changes the format and messes it up. I want to emphasize this point, since it could be that the patient has the old version of the pills, and what I try to do is avoid them taking the same pill twice.”*
			Doctor	*“There is one thing that works very well, at least in Sanlúcar de Guadiana, which is the pillbox for the elderly that is prepared by the pharmacist; the pharmacist gives the full pillbox to the patient each week, then the patient hands back the pillbox for a refill. I think we have to sign a document; this is being done with some patients in Sanlúcar, and the system works well. We have a close working relationship with this pharmacist; sometimes he even calls us about medication reconciliation issues.”*
	Visits	2	Nurse	*“Especially home visits. If necessary, we go daily.”*
			Doctor	
	Review of medicine cabinets	15	Nurse	*“As for control of medication, once every six months we review the medicine cabinets in my office. I call family members to review the medication they take, if there have been changes. This process is done by nurses based on the reports we have received from the hospital if you have been there, or on the latest reports issued by their GP. We ask that family members to know what the patient is taking that medication for, and that they understand why the patient needs to take it. We also ask them, and especially diabetics, about the adverse effects that medicines can cause, and above all, to maintain strict glycemic control. The truth is that the relatives seem to understand all this very well. Apart from that control, when I make home visits, I take the opportunity to check the pillboxes; for most patients we have their pillboxes with the medication for the day or the week; those pillboxes are prepared by the family before going to work, or by the home-help. This help is essential.”*
			Doctor	*“When they come from being discharged from hospital, they sometimes come to ask how they have to take their medication, or when the medication changes laboratory, then we also have to be aware and explain that they are the same medicines but in a different shape.”*

## Data Availability

Not applicable.

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
