# Peer review of "Influence of the COVID-19 Pandemic on Medication Reconciliation in Frail Elderly People at Hospital Discharge: Perception of Healthcare Professionals"

_ijerph, 2022, doi:10.3390/ijerph191610348_

Round 1

Reviewer 1 Report

Medication reconciliation in frail older adults at hospital discharge is often a problem. This problem has unquestionably increased during the COVID-19 pandemic. The article highlight how doctors and nurses lived through this situation and what difficulties they encountered. I will give you some suggestions to improve your paper.

Title: "Professional perception"... It will be better if you use "nurses and doctors..."

 Line 17, You talk about a pandemic, you should concretize for the COVID 19 pandemic.

Material and methods: You must accurately describe the selection of participants as well as the location and context of the interviews. Line 160, "...Influence of Covid-19...": Not the influence of the disease itself, but the context experienced due to the pandemic. Modify, please.

Why did you use a phenomenological approach to get to know perceptions and not lived experiences? Why did you use a phenomenological approach with theoretical sampling?

The coding of the interviewees' arguments generated 41 codes, divided into four groups, but you only present three of them. Which group did you decide to despise?

Is the quantification of citations relevant in phenomenology, or even coherent? You must justify this option.

The ethical aspects go far beyond informed consent and must all be made explicit.

You consider the virtual interview as a limitation. The virtual realization of the interview needs to come in the method.

On line 207 please correct the lapse "inter-views"; on line 222 correct "con-sequences" and on line 223 put full stop. One line 318 please correct "professionals65and close", on line 349 correct "cli-mate" and on line 364 please correct "in-volved".

There are some formatting issues with the first column of the tables and on page 9 it is still necessary to put a "p" in the word "Phone".

Author Response

REVIEWER1

Medication reconciliation in frail older adults at hospital discharge is often a problem. This problem has unquestionably increased during the COVID-19 pandemic. The article highlight how doctors and nurses lived through this situation and what difficulties they encountered. I will give you some suggestions to improve your paper.

Title: "Professional perception"... It will be better if you use "nurses and doctors..."

Thank you for this pertinent advice, with the modification we have made to the title it is more attractive and in line with the content (2-4 line). Thanks a lot.

Line 17, You talk about a pandemic, you should concretize for the COVID 19 pandemic. (DONE).

Thank you very much, we have modified it, line 17.

Material and methods: You must accurately describe the selection of participants as well as the location and context of the interviews. Line 160, "... Influence of Covid-19...": Not the influence of the disease itself, but the context experienced due to the pandemic. Modify, please.

Thank you very much for your corrections, we have modified the methodology section that describes the procedure in more detail (138-170 line).

 we have specified that the context generated by the pandemic is the one that has influenced professionals as you told us. 133-134 line

Why did you use a phenomenological approach to get to know perceptions and not lived experiences? Why did you use a phenomenological approach with theoretical sampling?

Qualitative research captures social reality through humane eyes, though an ample, flexible and profound approach. Inside, descriptive phenomenology accounts on human experiences of perceived phenomena, told in first person and presented as structures of meaning. This has turned into an important contribution to the ways of thought and development of health professional,  therefore, there is an insight in this paper about to know the perception of primary care professionals of the influence of the Covid-19 pandemic on the medication reconciliation process in patients over 65 years of age after hospital discharge.  We know through this approach the human experience of how professionals have perceived this phenomenon.

We have to thank you for your comments, since we have noticed an error in the wording of the sample selection, since the sampling carried out was an intentional sampling in the study areas, rural and urban, selecting informants who have lived the experience the influence of the Covid-19 pandemic on the medication reconciliation process in patients over 65 years of age after hospital discharge. We have modified this error in the manuscript, in line 139-143

The coding of the interviewees' arguments generated 41 codes, divided into four groups, but you only present three of them. Which group did you decide to despise?

Thank you very much for your appreciation, there has been a typographical error, there were not four groups but 3 groups generated for the analysis.

Is the quantification of citations relevant in phenomenology, or even coherent? You must justify this option.

Data analysis in phenomenology impliesat the time of research a mixture of phenomenological reduction (predisposition to be surprised by the data leaving one's own "assumptions/ideas" aside, bracketing), analysis, intuition and description.

The hermeneutical circle includes isolation of case-paradigms, identification of repetitive themes in the discourse , and selection of exemplary quotations to illustrate themes. The counting of the citations has allowed us to simplify this analysis process, being able to quantify the repetitive topics in the discourse of the informants and how dense some codes were (for example conciliation tools) facilitating the process of analysis, intuition and description.

The ethical aspects go far beyond informed consent and must all be made explicit.

Thank you very much for your comments so successful, we have expanded this section (207-214 line)

You consider the virtual interview as a limitation. The virtual realization of the interview needs to come in the method.

Interviews are part of the qualitative work and is an effective tool to unravel meanings elaborated by the subjects in their speeches, stories and experiences. The social distance resulting from the COVID-19 pandemic strengthened the already emergent process of virtual connections between people, bringing implications also for conducting research.

I attach the bibliographic reference on a systematic review that aimed to assess the reliability of the use of online interviews for health research (Schmidt, B., Palazzi, A., & Piccinini, C. A. (2020). Online interviews: potentialities and challenges for dice ponytail no context gives COVID-19 pandemic. Revista Família, Ciclos de Vida e Saúde no Contexto Social, 8(4), 960-966), these authors conclude "like all data collection strategies, online interviews have advantages and disadvantages. Despite the challenges, it is understood that online interviews have potential, especially with regard to qualitative studies in the context of COVID-19, because it is one of the few alternatives to access real scenarios with geographical dispersion in a short period of time, in addition to the possibility of investigating various topics present in the lives of individuals and families in social distancing"

We have included in the methodology section, in the data collection process that we have used this type of interview, thank you very much for your accurate appreciation (180-182 line).

On line 207 please correct the lapse "inter-views"; on line 222 correct "con-sequences" and on line 223 put full stop. One line 318 please correct "professionals65and close", on line 349 correct "cli-mate" and on line 364 please correct "in-volved". (DONE)

There are some formatting issues with the first column of the tables and on page 9 it is still necessary to put a "p" in the word "Phone". (DONE)

Reviewer 2 Report

Appreciate the authors of this manuscript I’m looking at an important problem, medication reconciliation, and the impact of COVID-19 on this important process. I do have some recommendations which may help this manuscript become more succinct, sound, and a stronger read to interested parties:

1.  Introduction — I find the introduction is quite long and can definitely be condensed and flow better. I think the information there overall is sound but definitely should flow from the general concept of aging patients and medications being a major problem in this group to the use of medication reconciliation in the past in order to address this problem to the question of how COVID-19 affected this important process. Right now your introduction  is twice as long as your discussion; I think a lot of the introduction could be condensed to get the idea across without dragging on some of the concepts, and some of the referenced information in the introduction can be moved to the discussion to back up your learning points.

2. Methods — I think this needs to be cleaned up and re-ordered. The first section should be to your patient setting and population. Describe in detail how you chose the physicians and nurses you studied, where they practice and what they do. Add in your study time frame here and name off if you were approaching a larger population and these are the only ones that wanted to participate with your questionnaire or did you only approach these folks? (Inclusion/exclusion criteria for the questionnaire). It is an option to put IRB/ethics acceptance here and consent.  The second section should discuss your study design. What were the actual questions asked and was this just a freeflowing open form discussion or were specific questions asked. How did you come up with this methodology for The questions ask we put in here. I don’t think you need the long quotation from Mottier in the method section. The last section would be your data collection and statistical analysis. What you have down should fit into this section nicely.

3. Discussion— I think the section could be beefed up a little bit including some of the data and information from your introduction. Also a focus on more solutions based off of your found data would be helpful. As this is just a descriptive study, it’s hard to weed out what is a true problem or a perceived problem by practitioners. As telemedicine plays a big role in the COVID-19 pandemic as well as the future of medicine (it’s not going away for sure) how does medication reconciliation take this into affect and drive for the better process to address providers concerns. Also, is there any information from the patient side on their perceived interactions and communications with providers especially pertaining to medication reconciliation.

4. Limitations— I would definitely include that this is a questionnaire with descriptive and qualitative analysis; I think you’re analysis overall is sound but the limitation is that you don’t have any real data presented that shows that there was an increase in medication errors or an Analysis from the patient side showing time spent decreased or medication problems increased. It’s not that surveys with qualitative analysis are not useful; they definitely have their place and can help drive practice change. You just have to acknowledge if they’re not as strong as a cohort analysis of data points. So please add that in.

Author Response

REVIEWER 2

Appreciate the authors of this manuscript I’m looking at an important problem, medication reconciliation, and the impact of COVID-19 on this important process. I do have some recommendations which may help this manuscript become more succinct, sound, and a stronger read to interested parties:

  1. Introduction — I find the introduction is quite long and can definitely be condensed and flow better. I think the information there overall is sound but definitely should flow from the general concept of aging patients and medications being a major problem in this group to the use of medication reconciliation in the past in order to address this problem to the question of how COVID-19 affected this important process. Right now your introduction is twice as long as your discussion; I think a lot of the introduction could be condensed to get the idea across without dragging on some of the concepts, and some of the referenced information in the introduction can be moved to the discussion to back up your learning points.

Thank you for your advice, we have condensed the introduction as you suggest, in order to make it more attractive and fluid for the reader respecting the main concepts. At the same time, we have highlighted the drug problems in the elderly, the importance of conciliation and the influence of the Covid-19 pandemic on this process.

On the other hand we have transferred part of the information and references to the discussion to give it more consistency following your indications.

Remaining as follows;

  • Elimination lines 72 to 79: The paragraph results in the idea of diagnostic errors and the concept is sufficiently clear by suppressing it.
  • Elimination of lines 85 to 99, moving part of the content to the Discussion section
  • Lines 119 to 126: deleted and transfer of content to discussion
  • Lines 127 to 129: Deleted and replaced by lines 97 to 102
  • Lines 140 to 144 deleted for resulting in the idea of the new model of care during the COVID-19 pandemic and transferred the content to the discussion

  1. Methods — I think this needs to be cleaned up and re-ordered. The first section should be to your patient setting and population. Describe in detail how you chose the physicians and nurses you studied, where they practice and what they do. Add in your study time frame here and name off if you were approaching a larger population and these are the only ones that wanted to participate with your questionnaire or did you only approach these folks? (Inclusion/exclusion criteria for the questionnaire). It is an option to put IRB/ethics acceptance here and consent. The second section should discuss your study design. What were the actual questions asked and was this just a freeflowing open form discussion or were specific questions asked. How did you come up with this methodology for The questions ask we put in here. I don’t think you need the long quotation from Mottier in the method section. The last section would be your data collection and statistical analysis. What you have down should fit into this section nicely.

Thank you very much for your appreciations, we have modified this section to improve reader understanding. (138-170 line)

Regarding the ethical aspects that you refer to, we have not included it in this section since the format of the IJERPH for the articles requires that it be described in another section.

We have removed the quote from mottier as it suggests and we have noticed that this section is more attractive to the reader after the modifications that you have described to us.

  1. Discussion— I think the section could be beefed up a little bit including some of the data and information from your introduction. Also a focus on more solutions based off of your found data would be helpful. As this is just a descriptive study, it’s hard to weed out what is a true problem or a perceived problem by practitioners. As telemedicine plays a big role in the COVID-19 pandemic as well as the future of medicine (it’s not going away for sure) how does medication reconciliation take this into affect and drive for the better process to address providers concerns. Also, is there any information from the patient side on their perceived interactions and communications with providers especially pertaining to medication reconciliation.

Thank you very much again for your comments, we have included as you refer in your comments data and information that were developed in the introduction to reinforce the discussion. Regarding the comment related to the need to have information on the perception of patients, we agree that it would be valuable and complementary information to the study, however, as we have mentioned in the introduction section, probably due to the recent Covid-19 phenomenon, we have not found studies that measure the reconciliation of medication in the context of the pandemic from the point of view of patients, although if a study that collects perceptions of professionals and that we have introduced cites in the aprtado of discussion.

 From line 308, we have changed wording and expanded content from the introduction, in addition to inserting two new bibliographic references.

  1. Limitations— I would definitely include that this is a questionnaire with descriptive and qualitative analysis; I think you’re analysis overall is sound but the limitation is that you don’t have any real data presented that shows that there was an increase in medication errors or an Analysis from the patient side showing time spent decreased or medication problems increased. It’s not that surveys with qualitative analysis are not useful; they definitely have their place and can help drive practice change. You just have to acknowledge if they’re not as strong as a cohort analysis of data points. So please add that in.

Thank you for your appreciation, we didn't really present any real data showing that there were medication errors, because the goal of the research was know the perception of primary care professionals of the influence of the Covid-19 pandemic on the medication reconciliation pro-cess, qualitative data are real, even if they are not numbers, they are speeches that show us the depth of the study phenomenon, which in this case was not to describe how many errors or show the analysis from the patient side showing time spent decreased or medication problems increased. The qualitative method is manifested in its strategy to try to know the facts, processes, structures and people in their entirety, and not through the measurement of some of its elements. The same strategy already indicates the use of procedures that give a unique character to the observations, so it is not a question of recognizing whether or not they are stronger than a cohort analysis of data points, it is simply the necessary strategy to respond coherently to the proposed objective.

We have added the limitation of being an online interview, another limitation would be to extrapolate these results to a different population, but we have really become aware of it and we have not described it.

Thanks again for your comments.

Round 2

Reviewer 2 Report

I thank the authors for their thorough review, rewrite, and additions that they have made to the manuscript. I think that these changes are a vast improvement and that the manuscript both reads much smoother than the previous version and also delivers a stronger message in the end. I have no other additions to be made at this time; thank you for your hard work.